# Application of Biocompatible Drug Delivery Nanosystems for the Treatment of Naturally Occurring Cancer in Dogs

**DOI:** 10.3390/jfb13030116

**Published:** 2022-08-07

**Authors:** Nicola Ambrosio, Silvia Voci, Agnese Gagliardi, Ernesto Palma, Massimo Fresta, Donato Cosco

**Affiliations:** Department of Health Sciences, University “Magna Græcia” of Catanzaro, Campus Universitario “S Venuta”, I-88100 Catanzaro, Italy

**Keywords:** cancer, dogs, liposomes, nanoparticles

## Abstract

Background: Cancer is a common disease in dogs, with a growing incidence related to the age of the animal. Nanotechnology is being employed in the veterinary field in the same manner as in human therapy. Aim: This review focuses on the application of biocompatible nanocarriers for the treatment of canine cancer, paying attention to the experimental studies performed on dogs with spontaneously occurring cancer. Methods: The most important experimental investigations based on the use of lipid and non-lipid nanosystems proposed for the treatment of canine cancer, such as liposomes and polymeric nanoparticles containing doxorubicin, paclitaxel and cisplatin, are described and their in vivo fate and antitumor features discussed. Conclusions: Dogs affected by spontaneous cancers are useful models for evaluating the efficacy of drug delivery systems containing antitumor compounds.

## 1. Introduction

Cancer is the leading cause of death in dogs over one year of age and has an incidence three times higher than that of traumatic injury [1]. Fifty percent of dogs above ten years of age develop cancer, and 25% of them die due to this neoplasm [2]. Like humans, dogs have a lifetime risk of cancer between 25 and 50%, but due to their shorter lifespans, they have an annual incidence of cancer up to 10-fold higher than that of humans. In fact, in the United States between 4 and 6 million new cases are diagnosed each year in a population of just under 90 million animals [3]. Non-Hodgkin lymphoma, malignant melanoma, osteosarcoma, and bladder carcinoma are the types of tumors commonly diagnosed [4]. It has been demonstrated that there are differing incidences of the various types of cancer depending on the breed of the dog. Canine cancer occurs spontaneously and shares a similar pathophysiology and clinical manifestation with that of human analogs. In addition, genomic analyses demonstrate that some genetic similarities exist between human and canine cancer. For these reasons, canine tumor models could help researchers to better understand important human cancer pathways and also to develop more reliable scientific testing [5]. A more detailed description of cancer’s causes, pathways, features and treatment are well described in several remarkable reviews [6,7,8,9].

Surgery, chemotherapy, and radiation similar to those used in human cancer treatment are the most important and implemented therapeutic approaches for canine malignancies. Surgery and radiation therapy are gold standards for treating primary local or completely resectable tumors, while chemotherapy is the only therapeutic approach for the treatment of advanced non-operable, metastatic, or recurrent malignant tumors [10]. Nowadays, several different chemotherapeutics are available. These drugs have many side effects that can compromise their correct use, and sometimes the pharmacological treatment must be interrupted [11]. Nanotechnology could contribute significantly to solving some of these problems. In fact, the entrapment of a molecule within nanoparticles is able to modify its pharmacokinetic and biodistribution profiles, improving the efficacy of the molecule while at the same time decreasing the side effects [12]. Moreover, thanks to their unique characteristics such as nanometric sizes, high surface-area-to-volume ratio, targeting features and the opportunity to exploit the enhanced permeability and retention (EPR) effect when systemically administered for the treatment of solid tumors, nanocarriers have been proposed as innovative systems to be used in anticancer therapy (Figure 1) [13]. A comprehensive description of this topic is beyond the aim of this review; however, a number of excellent articles has been already published [14,15,16].

The aim of this review is to discuss the state-of-the-art use of nanoparticles for the treatment of dogs’ tumors and how the canine tumor model could be useful for the development of innovative nanomedicine.

## 2. Comparative Oncology: How the Canine Model Can Help Research in Cancer Treatment

Comparative oncology is the study of spontaneously-occurring cancer across different animal species with the aim of favoring advancement for the benefit of human and animal health [17]. In fact, establishing tumor-bearing animals as complementary models of human cancers gives researchers the opportunity to discover and understand unknown molecular mechanisms in order to propose novel treatments for humans [18,19].

For these reasons there is much sustained interest focused on dogs, based on the increasing number of cancers diagnosed every year [20]. In addition, other aspects such as a similar body size (some of the larger dogs can be over 100 kg), an environment shared with humans, a similar immune system, and the types and spontaneity of cancer make dogs an attractive complementary model. Canine cancer shares more similarities with human tumors than do the traditional models being used to investigate these diseases such as zebrafish, nematodes, rodents, or frogs. These are all characterized by a shorter lifespan and a different, more rapid typology of cancer development, but dogs spontaneously develop various forms of cancer which share similar biological features with their human counterparts [21]. Some oncogenes and tumor suppressor genes which play an important role in human cancers are also involved in canine cancers. Many canine tumors, in addition, spontaneously develop metastases and this could help in the discovery of unknown correlated features and furnish the opportunity to realize a better comparison model [22]. 

Moreover, since dogs share the same environment with humans, they are exposed to the same external influences and risk factors for developing similar types of tumors [23]. The same types of cancer are diagnosed both in humans and dogs exposed to the same carcinogenic agents, as demonstrated by various epidemiologic studies. This is true, for example, in several industrialized regions of North America where, due to exposure to the same carcinogenic pollutant, a high incidence of bladder cancer both in humans and in dogs was registered. Interestingly, the latent period was much shorter in dogs (less than ten years) than in humans (above twenty years), suggesting that dogs could be considered sentinels providing an early identification of carcinogenic agents [24].

## 3. Drug Delivery Systems for the Treatment of Canine Tumors

### 3.1. (Phospho) Lipid-Based Nanosystems

#### 3.1.1. Liposomes

Liposomes are spherical vesicles made up of natural or synthetic phospholipids characterized by one or several inner aqueous compartments surrounded by one or more bilayers, and hence having the same supramolecular organization as living cells. Considering their very low toxicity, wide versatility and their peculiar physico-chemical properties, liposomes are the most extensively-studied drug delivery systems [25,26]. Their structure has various advantages which include: (i) allowing the encapsulation of active compounds characterized by different physico-chemical properties (also as a multidrug carrier); (ii) preserving drugs from destabilization processes; (iii) decreasing the side effects of the entrapped compound(s) as a result of a lower efficacious dosage; and (iv) intravenous (IV) modulation of the pharmacokinetic profiles of the active compounds [27,28]. It is also possible to modify the lipid composition in order to alter the rigidity of the bilayer, the phase transition temperature and physical stability, all of which can influence the drug retention rate and its release [29]. In this context, the coating of their surfaces with hydrophilic polymers or specific derivatives has been shown to increase their stability and half-lives after IV administration, thus promoting their localization in several solid tumors. For instance, the PEGylation of liposomes avoids their opsonization and uptake into the macrophagic-phagocytic system [30,31]. Even though the liposomes are safe and nontoxic, they can promote acute hypersensitivity reactions related to the activation of the complement system as a consequence of their surface architecture [32].

There are essentially two ways exploited by liposomes for reaching solid tumors. One is a passive targeting that promotes the localization of the vesicles inside the tumor by exploiting the patho-physiological conditions of the tissues, the peculiar architecture of the blood vessels and the so-called enhanced permeability and retention effect [33]. The other is an active targeting in which different ligands can be linked to the liposomal surface in order to promote their interaction with the receptors that are overexpressed in tumor tissues [34]. A third method associated with the previous ones can be also considered and is based on the use of parameters or external stimuli such us temperature, pH or magnetic fields for increasing the release of the entrapped compounds from the liposomes directly into the tumor area [28]. 

Doxil was the first liposomal formulation containing an anticancer drug (doxorubicin hydrochloride) approved by the Food and Drug Administration (FDA) in 1995 for pharmaceutical application [35]. Doxil is currently used for the treatment of various human cancers [33]. Successively, other liposomal carriers for the delivery of anticancer drugs, such as Depocyt or Marqibo containing cytarabine and vincristine, respectively, were approved [36]. Various in vitro and in vivo studies have been carried out in order to evaluate the activity and/or efficacy of lipid-based anticancer formulations on different canine cancer cells and their pharmacokinetics and/or safety. These studies are summarized in Table 1.

Very few in vivo studies on dogs with spontaneously occurring tumors have been performed to evaluate the safety, efficacy, and pharmacokinetic parameters of liposomal anticancer drugs. However, in this review we focus more on this kind of study. Many of these in vivo studies have been based on the use of nanopharmaceuticals containing doxorubicin for the treatment of various canine tumors. The first evidence in which a dog with a spontaneously occurring neoplasm was treated with liposomal doxorubicin was described by Kisseberth et al. in 1995 [52]. In detail, a 12-year-old female dog with multiple myeloma had already been treated with conventional chemotherapy (melphalan and prednisone firstly, then associated with vincristine after seventeen weeks and finally free doxorubicin after five additional weeks) which resulted in only a partial response. The dog then received the active compound contained in liposomes by IV injection at a drug concentration of 35 mg/m^2^ every three weeks for six cycles combined with melphalan. Despite a conventional chemotherapy treatment of 41 weeks without great therapeutic benefits, the animal evidenced a decrease in the IgAs and neutrophils at the beginning of the treatment with liposomal doxorubicin, which remained constant over the next weeks. Liposomal doxorubicin induced a durable, complete response without any apparent clinical cardiotoxicity. Moreover, this study also demonstrated there was no appearance of drug resistance [52]. An interesting and more extended trial was conducted on fifty-one dogs suffering from different forms of tumors in order to evaluate the safety and efficacy of Doxil administrated by IV injection at a drug concentration of 0.75–1.1 mg/kg every three weeks [53]. No myelosuppression and less gastrointestinal toxicity were observed compared with the dogs treated with the free form of doxorubicin hydrochloride. Moreover, the cutaneous side effects, characterized by mild erythema, hyperemia, edema, alopecia, and in some cases severe crusting, ulceration, and epidermal necrosis were considered dose-limiting toxicity. These severe cutaneous lesions, which closely resembled the palmar-plantar erythrodysesthesia (PPES) that normally occurs in humans, were self-limiting and typically resolved in 1–2 weeks. The results demonstrated the efficacy of Doxil with an overall response rate of 25.5%, including five complete and eight partial remissions [53]. Even though this data was very encouraging, additional investigations were needed with the aim of evaluating the real efficacy of chemotherapeutic liposomal formulations for use on dogs. In this context a more specific study was performed by Teske et al. that evaluated the toxicity and efficacy of Doxil for the treatment of splenic hemangiosarcoma [54]. Two groups of ill dogs received two different treatments within three weeks after surgery: the first one was treated with 20 mg/m^2^ of Doxil once every three weeks for six total treatments, and the other group received 30 mg/m^2^ of free doxorubicin for the same period. The results did not evidence any significant difference in the average disease-free and the average overall survival rates between the two different treatment groups. On the other hand, it is interesting to note the incidence of adverse effects. In fact, fewer cases of anorexia, emesis, neutropenia and thrombocytopenia resulted in the dogs treated with Doxil compared with those that received the free drug [54]. In the same context, Sorenmo et al. carried out a clinical trial to evaluate the efficacy of PEGylated liposomes containing doxorubicin hydrochloride administered by the intraperitoneal route for the prevention of intra-abdominal tumor recurrence in dogs with hemangiosarcoma [55]. Fourteen dogs with splenic hemangiosarcoma were treated with a dosage of 1 mg/kg of liposomal doxorubicin injected every three weeks for four treatments. The treatment, unfortunately, was ineffective; in fact, twelve of the fourteen dogs died due to hemangiosarcoma-related causes, hepatic metastasis and hemoabdomen, and two died from other causes. These negative clinical outcomes could be related to the excessively low mesenteric concentration of the drug that resulted from the treatment [55].

Finally, as is well known, cutaneous reactions, especially PPES, could be dose-limiting for Doxil [56]. Vail et al. performed a trial on forty-one dogs with non-Hodgkin’s lymphoma with the aim of evaluating the efficacy of pyridoxine while decreasing the cutaneous toxicity associated with Doxil [57]. The dogs received 1 mg/kg IV of Doxil every three weeks, and were randomly divided into two groups (a pyridoxine group and a placebo group). The pyridoxine group received, in addition to the chosen chemotherapeutic agent, pyridoxine at a dosage of 50 mg/kg by oral administration three times a day for fifteen weeks, while the placebo group received lactose on the same schedule. The results showed a 4.2 times higher likelihood of developing PPES for the dogs in the placebo group compared with those treated with pyridoxine. This outcome corresponded to a significant difference in the maximum cumulative dose, which was 4.7 mg/kg for the pyridoxine group and 2.75 mg/kg in the placebo group. Finally, pyridoxine did not influence the pharmacological response to Doxil; in fact, remission rates were similar in both groups [57]. 

Although in fewer numbers, other chemotherapeutic agents encapsulated in liposomal formulations have been also tested. In one trial conducted by Vail et al., a stealth liposome containing cisplatin was compared with free carboplatin as adjuvant therapy subsequent to surgery in the treatment of osteosarcoma [58]. The dogs were divided into two groups receiving the different formulations seven days before surgery. The first group intravenously received the liposomal cisplatin at a drug concentration of 350 mg/m^2^ (five times the established maximum tolerated dose), while the second one was treated intravenously with 300 mg/m^2^ of carboplatin (less toxic with respect to cisplatin in the free form) [59]. Both treatments were repeated every three weeks for a total of four times. The results demonstrated that the encapsulation of the active compound within the vesicular carrier did not enhance its pharmacological efficacy, as shown by the average disease-free and OS time, but it promoted a better long-term survival rate of the dogs compared with the animals treated with carboplatin. In fact, eight of the nine dogs remained alive with remission of the disease. Even though this study provided encouraging data, additional research is necessary due to the small number of animals studied [58]. 

A very interesting study was performed using highly positively-charged liposomes containing untargeted tumor RNA (RNA-NPs) to treat a dog with malignant glioma [60]. Once the potential in vitro cytotoxicity of the liposomal formulation had been ascertained, it was administered to the tumor-bearing dog on a weekly basis three times. The systems containing RNA proved to be safe, as demonstrated by the results obtained from blood counts, liver, and renal tests, and elicited an increase in serum IFN-μ, PD-L1, CD80, CD86, and MHCII on CD11c^+^ six hours post injection. The dog was in a stable condition at the end of the treatments, showing consistent tumor progression or pseudoprogression without needing surgical resection or radiotherapy [60]. 

Kamstock et al. performed a study on thirteen cutaneous tumor-bearing dogs in order to determine whether the use of liposome–DNA complexes (LDC) could modulate gene expression and influence angiogenesis [61]. The dogs received two different types of LDC: seven dogs were treated with LDC-containing canine endostatin (an endogenous angiogenesis inhibitor), while the other six dogs received a liposomal formulation containing a DNA-encoding luciferase. Both LDC treatments were proven to be safe and had only transient, mild side-effects such as lymphopenia 24 h after the first administration. The LDC treatments inhibited tumor growth in eight dogs, induced tumor regression in two dogs—one reached a complete response and one a partial response –, whereas only three dogs showed an increase in the tumor mass. Interestingly, only slight differences emerged between the two LDC treatments, demonstrating that LDC can promote an inhibition of angiogenesis independently from transgenic delivery [61]. 

Remarkable studies based on the administration of liposomal muramyl tripeptide-phosphatidylethanolamine (L-MTP-PE or Mifamurtide) in dogs were carried out in the late 1990s with the aim of assessing its efficacy and safety as adjuvant to the formulation in the treatment of various kinds of tumors [62,63,64,65,66]. Mifamurtide obtained the EMA’s marketing authorization in 2009 and is currently used to treat human high-grade non-metastatic osteosarcoma [67]. One trial was performed on dogs with malignant and highly metastatic spontaneous cancer –hemangiosarcoma and osteosarcoma—in order to evaluate the safety and efficacy of L-MTP-PE in association with traditional chemotherapeutic drugs (doxorubicin, cyclophosphamide or cisplatin) [62]. At the end of the treatment, the dogs randomly received 2 mg/m^2^ L-MTP-PE or a placebo, a saline liposomal solution characterized by the same lipid composition as L-MTP-PE, twice a week for eight weeks. Two separate trials were carried out based on the kind of tumor, i.e., hemangiosarcoma or osteosarcoma. All twenty-seven dogs of the hemangiosarcoma trial, which had been first surgically treated with splenectomy, received IV doxorubicin and cyclophosphamide every three weeks for a total of four treatments, at a drug dosage of 30 mg/m^2^ and 100 mg/m^2^, respectively. Twelve of them received L-MTP-PE, while the other fifteen received a placebo. Fifty percent of the dogs treated with L-MTP-PE died due to metastasis and an average survival time of 9.1 months was reached compared with 73% deaths due to metastasis with an average survival time of 4 months registered in the placebo group. The dogs in the osteosarcoma trial were initially treated with surgical removal of the primary tumor and then received 70 mg/m^2^ of cisplatin once every four weeks for a total of four times. Eleven of them received L-MTP-PE with the same modalities as those adopted in the hemangiosarcoma trial, while the other fourteen received a placebo. In this case also, the average survival rate of the dogs treated with L-MTP-PE was longer than that registered for the placebo group (14.4 months vs. 9.8 months, respectively). Another important and interesting outcome was that 27% of the animals treated with L-MTP-PE had a survival time greater than two years. Moreover, another trial was carried out on dogs with spontaneous osteosarcoma treated with L-MTP-PE adopting a different protocol to that in the aforementioned study. In this trial, in fact, the treatment with L-MTP-PE began simultaneously with the administration of cisplatin [63]. In detail, all the dogs received 70 mg/m^2^ of cisplatin every three weeks for a total of four times after the surgical procedure. One day after the first administration, the sixty-four dogs of the study were randomly divided into three groups in order to receive 2 mg/m^2^ L-MTP-PE either once or twice a week, or a placebo once a week, for a total of eight weeks. There were slight differences in the average of metastasis-free intervals among the three groups (7.5, 6.3 and 5.8 months for twice L-MTP-PE, once L-MTP-PE and placebo, respectively). Neither were there any large differences in the average recurrence times of the metastases among the three groups (10.3, 10.5 and 7.6 weeks, respectively). Nineteen of the twenty-one dogs (90%) treated twice a week with L-MTP-PE developed metastases and seven (33%) of them survived for more than one year. Eighteen of the twenty-one dogs (86%) treated once a week with L-MTP-PE were characterized by metastasis and seven of them (33%) survived for over a year. Remarkably, in the placebo group, seventeen animals of the twenty-two (77%) developed metastases and nine dogs (41%) survived more than a year [63]. 

Another study was also performed on dogs with hemangiosarcoma treated post-splenectomy with 2 mg/m^2^ of L-MTP-PE or a placebo in association with 30 mg/m^2^ of doxorubicin and 100 mg/m^2^ cyclophosphamide [64]. The results confirmed the efficacy of L-MTP-PE; in fact, 37% of the dogs treated with L-MTP-PE were still alive one year later, compared with 15% of the placebo group [64]. Moreover, the antitumor activity of L-MTP-PE was also tested on dogs with mammary carcinoma [65], resulting in no significant anticancer effects when used as adjuvant. Lastly, a different and more extended study was performed in order to assess the antitumor efficacy of L-MTP-PE as adjuvant in melanoma therapy [66]. Ninety-eight dogs with oral melanoma were divided into two groups in order to carry out two trials: in the first, fifty dogs were classified as a function of the clinical stage of the disease and, after surgery, were randomly treated with L-MTP-PE once a week for eight weeks or with a placebo; in the second trial, forty-eight dogs were classified by the clinical stage of the disease and the extent of surgery and then randomly treated with L-MTP-PE twice a week in addition to recombinant canine granulocyte macrophage colony-stimulating factor (rcGM-CSF) administered daily for nine weeks at a dosage of 15 µg/kg or with saline. In trial 1, the L-MTP-PE treatment revealed itself as being safe, and the efficacy was correlated to the clinical stage of the disease. In fact, the mean survival time was significantly higher in the dogs treated with L-MTP-PE than that of the animals receiving the placebo in the early clinical stage (stage I), but no significant differences resulted in the dogs with advanced clinical stages (stage II or stage III). In trial 2 no differences were detected between the dogs treated with L-MTP-PE and rcGM-CSF and those treated with the placebo, demonstrating the need to treat the animals in the early stage of the disease [66].

A specific experimental study was carried out by Hafeman et al. on five dogs with spontaneous malignant histiocytosis (MH) in order to assess the efficacy of a liposomal formulation containing clodronate [68]. In detail, firstly phosphatidylcholine-based liposomes were used to entrap the clodronate and then were tested in vitro on three different MH cells lines (DH82 and two other MH canine cells obtained from primary cultures of biopsies of tumor-bearing dogs). The analysis demonstrated that the cytotoxic effect of liposomal clodronate was greater than that of the free drug and that it manifested selective activity towards non-phagocytic tumor cells. The liposomal formulation was tested successively in vivo on five ill dogs, which gave heterogeneous results. In fact, only two of these five dogs evidenced a significant regression of the tumor mass. Interestingly, two of the three non-responding animals were Bernese Mountain dogs which could indicate a specific drug resistance related to the species, but obviously, additional studies are required [68]. 

In another study described by Withers et al., curcumin was encapsulated in a liposomal system (Lipocurc) [69]. As is well known, in fact, the major limitation of curcumin as a potential anticancer drug is related to its low degree of bioavailability due to its scarce solubility in aqueous media [70]. Lipocurc is a liposomal formulation of curcumin that enables the intravenous delivery of the drug, bypassing the problems associated with oral administration and the poor solubility of the active compound [71]. Withers et al. tested the anticancer activity of Lipocurc on canine cancer cells with the aim of proposing the formulation for the treatment of canine pulmonary neoplasia. The in vitro tests demonstrated that Lipocurc is characterized by antiproliferative activity similar to that of free curcumin, and so it is efficacious on various canine tumoral cell lines. An in vivo study was performed successively on nine dogs with primary pulmonary or metastatic neoplasia. The animals were administered 10 mg/kg of liposomal curcumin intravenously as a continuous infusion, once a week, for a total of four doses. The Lipocurc was quite safe and well tolerated when administered weekly in an 8-h infusion, but no dogs reached a complete regression of the disease using this dosage schedule. Moreover, the connection between Lipocurc and the death of two of the dogs could not be totally excluded. These results indicate that further studies are needed in order to assess the safety and efficacy of this formulation [69].

#### 3.1.2. Non Phospholipid-based Nanoparticles

In the ample field of lipid-based nanoparticles many drug delivery systems have been developed and characterized [72]. Various kinds of lipid derivates formulations have been employed and different structures have been obtained. As described for liposomes, some acute hypersensitivity reactions have been also described for these kinds of drug delivery systems [32,73,74,75]. Among these, one particular drug delivery platform was designed to contain water insoluble or quite soluble active compounds forming retinoid-based micellar formulations. This technological approach was used to develop Paccal Vet, a formulation containing paclitaxel proposed for veterinary application. Several studies have been performed with the aim of demonstrating the efficacy of Paccal Vet. For instance, in 2012 Vail et al. published a report describing the effects of the system on tumor-bearing dogs [76]. In detail, 252 dogs with advanced-stage, nonresectable mast-cell tumors were recruited in order to assess the safety and efficacy of Paccal Vet. The dogs were randomly divided into two groups and treated with 150 mg/m^2^ of micellar paclitaxel every three weeks for a total of four times or with 70 mg/m^2^ of lomustine used as control. The results confirmed that Paccal Vet has greater pharmacological efficacy and safety than lomustine [76].

Another experimental investigation was described by von Euler et al. that demonstrated the better in vivo efficacy of Paccal Vet (100–150 mg/m^2^) compared with the conventional formulation of paclitaxel (Figure 2) [77].

Taking advantage of the similarity between non-protein lipid nanoemulsions (LDE) and low-density lipoproteins, the receptors of which are overexpressed in cancer cells [78,79], an LDE containing carmustine was developed and its safety was compared with the free drug [80]. Fifty dogs with spontaneous lymphoma were recruited, and eight of them were treated with commercial carmustine, while the other seven dogs received carmustine encapsulated in LDE. The dogs of both groups received carmustine combined with vincristine and prednisone. The results showed that LDE-containing carmustine showed safety and efficacy profiles similar to those of the free drug, as demonstrated by the mean survival time (207 vs. 247 days, respectively), the mean progression-free interval (119 vs. 199 days, respectively) and the complete response (6 vs. 5 animals, respectively) [80]. 

All the lipid-based drug delivery systems previously discussed are summarized in Table 2 as a function of the type of tumor. 

### 3.2. Non Lipid Nanoparticles

Several drug delivery nanosystems made up of non-lipid materials have been developed in the past few decades that can be classified as a function of their physio-chemical properties. In order for them to be proposed for biomedical and pharmaceutical applications they need to be biocompatible, characterized by high stability in biological fluids and must have wide versatility allowing researchers to functionalize their structures and to modulate their degradation and interaction with various compounds. It is possible to obtain nanoparticles by means of several techniques and procedures in order to encapsulate different compounds; various reviews have already discussed these aspects, and therefore, they will not be described in this manuscript [12,13]. 

In this section, the experimental studies performed on dogs with spontaneously occurring cancer and concerning the application of non-lipid nanoparticles will be discussed, while other in vitro and in vivo investigations have been summarized in Table 3.

In the case of veterinary application, a trial was performed on thirteen dogs with heterogeneous, naturally-occurring cancers, such as anal sac, oral squamous cell, nasal or digital squamous cell carcinomas and oral melanoma, in order to evaluate the efficacy, safety and pharmacokinetics of a polymeric hyaluronan cisplatin-nanoconjugate (HA-Pt) with respect to the free compound [92]. Polymeric nanoparticles are classified as a function of the polymers used in their composition (natural or synthetic) or their physio-chemical properties (thermo-, pH-sensitive, hydrophilic, lipophilic, etc.) [93,94]. However, despite some advantages in terms of stability, safety and wide availability of the polymers as well as the opportunity to decorate the nanoparticles surface and to control the degradation, the interaction between polymeric nanoparticles and bloodstream components that cause physico-chemical changes of the nanoparticles, could limit the development and the pharmacological activity of polymeric nanoparticles [13,95]. The most important polymer-based formulation containing an antitumor compound is nab-paclitaxel (Abraxane^®^, Albumin nanoparticles containing the drug) that is approved for clinical application in humans for the treatment of several cancers [96]. In the mentioned study [92], the dogs received four doses at a drug concentration of 10–30 mg/m^2^ directly into the tumor or into the peri-tumoral submucosa every three weeks. The toxicity studies demonstrated that the encapsulated cisplatin was safer than cisplatin; in fact, HA-Pt did not decrease the ability of bone marrow to produce red cells and it did not reduce the number of platelets, but merely caused a small compromise in the synthesis of white blood cells. An important aspect is that HA-Pt did not promote any nephrotoxicity over time, which is the primary adverse effect of the active compound when administered in the free form, nor was the physiological renal activity greatly affected. The results confirmed that 23% of the animals had a complete regression of the disease, while 15% had a partial response [92]. Recently, Cai et al. developed a cisplatin-hyaluronan nanoconjugate (HylaPlat) with the aim of treating a dog with a non-resectable oral squamous cell carcinoma with metastases in the local lymph nodes [97]. The dog was treated with 5–10 mg/m^2^ of HylaPlat intra-lesionally injected every three weeks, for a total of four times. HylaPlat induced an important regression that favored the surgical resection of the tumor. In addition, the drug delivery system promoted the localization of cisplatin within the metastatic lymph nodes [98]. The dog reached a complete remission and maintained a stable condition for at least one year after the treatment (Figure 3) [97].

Another interesting polymer-based nanosystem was developed and tested in dogs with naturally-occurring osteosarcoma by Yin et al. [99]. In this case, polylactide nanoparticles containing doxorubicin and coated with pamidronate (Pam-Doxo-NPs) were synthetized, characterized and tested in vitro and in vivo. This was done in order to evaluate the binding feature of the nanosystems to hydroxyapatite and assess the antitumor activity and safety of the formulation on mice and in affected dogs. Following encouraging results obtained in vitro and in mice, nine dogs received Pam-Doxo-NPs as an IV infusion with an increasing concentration of the drug (0–180 mg/m^2^). The Pam-Doxo-NPs were rapidly distributed to major organs, such as the heart, kidney, liver, and spleen and were observed to accumulate in the tumor microenvironment after 1–2 h. Moreover, the Pam-Doxo-NPs were found to be safe even at the highest dosage used, and no hematic, renal, cardiac, or permanent liver toxicity occurred. Due to the similar tumor features between canine osteosarcoma and the human form (they are the most frequently-occurring primary bone tumors in children and adolescents), this study showed promising results even though additional and more extended investigations are required [99,100,101]. 

Another formulation was proposed by Young et al. to treat oral canine melanoma. In this study nanoparticles consisting of PEG, polylactide (PLA), and polycaprolactone (PCL) and containing temozolomide-loaded superparamagnetic iron oxide were directly injected into the tumor by means of CED [102]. This technique allows the delivery of drugs directly through the interstitial spaces of the central nervous system, bypassing the brain-blood barrier and increasing their concentration in the brain [103]. The use of magnetic iron oxide nanoparticles helps to avoid one of the limitations of this technique that is related to tracking the drug. It should also be noted that the stability of iron oxide nanoparticles in body fluids is a critical issue for future research [104,105]. The investigation was carried out on ten dogs, which received an infusion volume of ~250 mL of formulation containing an equivalent of 5 mg/kg of temozolomide by means of a sterile stepdown catheter. Magnetic Resonance Imaging demonstrated that the nanoparticles had localized within the tumor in seven out of ten dogs. Nine of the ten dogs demonstrated a full remission of the acute effects related to the procedure, confirming the safety of the nanosystems and the suitability of the technique for the treatment of glioma in a canine model. The results showed a significant decrease in the tumor mass in one dog with survival of over two years, while the average survival time of the other treated dogs was 72 days [102]. Hoopes et al. investigated the effects of radiotherapy on iron oxide nanoparticles and/or virus plant nanoparticles [106]. The latter have recently attracted great interest in the field of nanotechnology for their application of targeted drug delivery and immunotherapy [107]. The study was designed in order to treat five dogs with spontaneous oral melanoma using four different chemotherapeutic approaches: hypofractionated radiation therapy (HFRT), magnetic/iron oxide nanoparticle hyperthermia (mNPH), HFRT + virus-like nanoparticles (VLP), and HFRT + VLP + mNPH. All the dogs, expect for the animal that received only HFRT, showed a large degree of tumor regression. In addition, the dogs treated with HFRT + VLP and HFRT + VLP + mNPH remained tumor free until death [106]. In the field of metallic nanoparticles, gum arabic-coated radioactive gold nanoparticles (GA-AuNPs) were prepared and intra-lesionally injected in nine dogs with active prostate cancer [108]. Gold nanoparticles are characterized by a wide available surface area, tunable properties, and easy synthesis as well as plus biocompatibility, a good safety profile and particular optical properties that promote their use as theranostic systems. On the other hand, their high cost represents a significant drawback for their application in medical field [109]. Gum arabic is a water-soluble, biocompatible and biodegradable polymer, naturally extracted from the *Acacia* species and consisting of glycoproteins and polysaccharides, that is widely used as an emulsifier and stabilizer [110,111]. In the proposed study, the dogs received different amounts of GA-AuNPs as a function of breed, weight and tumor size. The post-injection body planar imaging evidenced that an average of 53% of the nanoparticles localized in the prostate. Only one dog showed severe acute local toxicity. The animals manifested no variation in their blood and weight parameters. Moreover, tumor regression was achieved in two of the dogs, while the other animals evidenced a stabilization of the tumor mass, which is a relevant result considering the fast development of this kind of disease [109].

All the non-lipid-based drug delivery systems previously discussed are summarized in Table 4 as a function of the type of tumor.

## 4. Conclusions

Nanotechnology applied to the field of medicine has made a large impact on the treatment of cancer-related diseases by promoting a decrease in the side effects of the drugs and increasing their pharmacological efficacy [112]. Along these lines, naturally-occurring canine cancer could help in the development of novel anticancer formulations. In fact, owing to their similarity to human cancer, the preclinical investigations and in vivo trials performed on dogs could facilitate advancement in the field of nanomedicine. Besides the studies described in this review, many other investigations are required for the translation of the most promising systems into daily practice, and at the moment, there are no drug delivery systems that have been approved for the treatment of cancer in dogs. This could certainly be due to the costs related to the development of innovative nanoformulations and experimentation. On the other hand, in many trials on dogs, nanoparticle-based formulations—as are more frequently used and documented in human trials—have failed and shown less encouraging results than the promising ones obtained after the initial in vivo tests. This could mainly be related to the differences between small animals, such as rats or mice, and bigger mammals such as dogs or humans. From this perspective, dogs with naturally-occurring neoplasia could represent a bridge between the murine models and humans, presenting the opportunity to save costs and optimize the innovative pharmaceutical formulations and protocols. Lastly, dogs share the same environment with humans and are sensible to similar carcinogens so they could be useful for the evaluation of these aspects. Dogs are nowadays considered an extension of the family, so finding a better anticancer therapy would be a great commercial opportunity for pharmaceutical companies as well as a “social” necessity, and nanomedicine can play an important role in reaching this goal. 

## Figures and Tables

**Figure 1 jfb-13-00116-f001:**
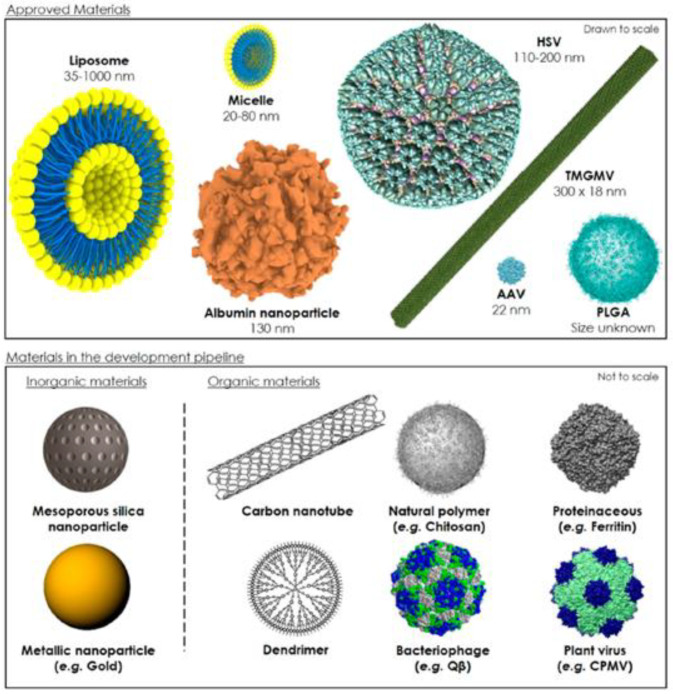
Schematic representation of the mean sizes and structure of approved and in development biomaterial-based nanocarriers. Reprinted with permission from [12]. Abbreviations. AAV: Adeno-associated virus; HSV: Herpes simplex virus; TMGMV: Tobacco mild green mosaic virus; CPMV: Cowpea mosaic virus.

**Figure 2 jfb-13-00116-f002:**
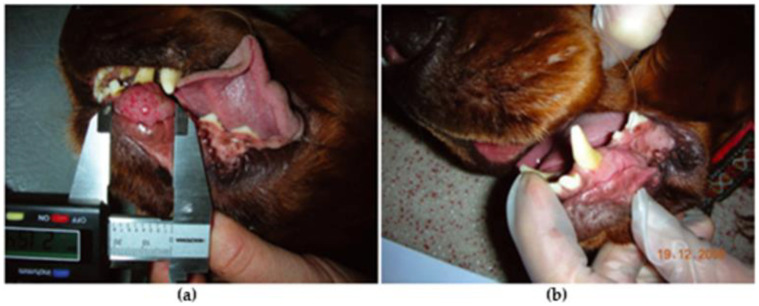
Treatment response of one dog with oral squamous cell carcinoma: (**a**) immediately prior to treatment; and (**b**) 21 days after cycle 2 of Paccal Vet treatment. Reproduced from [76].

**Figure 3 jfb-13-00116-f003:**
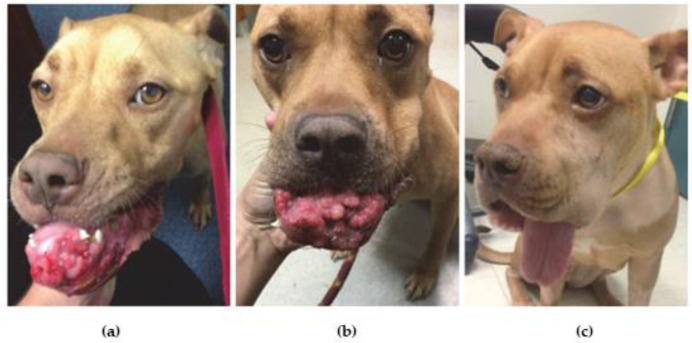
A dog with an inoperable oral squamous cell carcinoma with metastases in regional lymph nodes: (**a**) before the administration of HylaPlat; (**b**) after the second injection; (**c**) at the fourth injection of the formulation. Reproduced by [97].

**Table 1 jfb-13-00116-t001:** In vitro and in vivo investigations concerning the use of lipid-based drug delivery systems in veterinary application.

Drug Delivery System	Study Type	Obtained Results	Reference
Liposomes containing SN-38	Safety	Dogs were treated with three different dosages of liposomal SN-38 (0.4, 0.8 and 1.2 mg/kg of drug). The last one was considered the MTD ^1^, because emesis, decrease of hematopoiesis and neutropenia were registered in dogs treated with 1.2 mg/kg of SN-38. The other two dosages were well tolerated.	[37]
Low temperature sensitive liposomes containing doxorubicin	Safety and pharmacokinetic	Administration of liposomal formulation, followed by over 30 min of local tumor hyperthermia, was well tolerated by most subjects. The MTD ^1^ established was 0.93 mg/kg IV. Pharmacokinetic values resemble those of the free drug, except for clearance which was ~17-fold lower for the liposomal formulation. Doxorubicin’s intratumor concentrations were variable, probably as a consequence of the different tumor vascularization	[38]
Liposomes encapsulating topotecan using transmembrane NH_4_EDTA gradient	Pharmacokinetic	The encapsulation of topotecan within liposomes dramatically increases the plasmatic levels and decreases the plasmatic clearance. NH_4_EDTA-L’s AUC_0_ was 30-fold that of the free drug. Unexpectedly, NH_4_EDTA did not increase topotecan’s intraliposomal retention.	[39]
Liposomes containing paclitaxel	Pharmacokinetic	The liposomal formulation showed similar C_max_, a 2-fold lower AUC and half-time, and a 2-fold higher clearance and volume of distribution compared with the free form of paclitaxel after IV administration. Moreover, the concentration of liposomal paclitaxel was found to be higher in the lungs than in other organs	[40]
PEGylated liposomes containing topotecan	Safety	No skin toxicity was observed in healthy dogs after IV administration even when high concentrations of the drug were used.	[41]
Liposomes containing Aluminum-Chloride-Phthalocyanine	In vitro efficacy	Aluminum-Chloride-Phthalocyanine encapsulated within liposomes associated with LED light irradiation showed antineoplastic activity on canine mammary gland complex carcinoma cells.	[42]
Vincristine sulfate-loaded liposomes (Marqibo)	Pharmacokinetic	Marqibo significantly increases the AUC_0_ and C_max_ of the drug and drastically decreases its volume of distribution and clearance with respect to the free form of vincristine sulfate.	[43]
Liposomal paclitaxel (Lipusu)	Pharmacokinetic	Liposomal paclitaxel was quickly localized in various organs after IV administration, especially in the spleen and liver, but it was slowly eliminated.	[44]
Liposomal vincristine sulfate	Pharmacokinetic	Liposomal vincristine is characterized by an increased AUC_0_ and half-time and a decreased volume of distribution after IV administration in healthy beagles compared with free vincristine	[45]
Liposomes containing SN-38	Pharmacokinetic	The concentration of liposomal SN-38 quickly decreases after IV administration. The elimination profile is independent of the injected dose.	[46]
Multivesicular liposomes containing cytarabine	Pharmacokinetic	Liposomal cytarabine (LC) reaches a t_max_ 4-fold higher than free drug (FC) after subcutaneous administration. Cytarabine-loaded multivesicular liposomes did not reach the cytotoxic plasma concentration with respect to its free form after s.c. administration. Only 20–30% of the injected liposomes were absorbed. The elimination profiles of the two forms of the active compound were similar.	[47]
Temperature-sensitive liposomes containing doxorubicin	Biodistribution and safety	Temperature-sensitive liposomal doxorubicin increased the localization of the active compound in the brain when combined with 15–30 min local hyperthermia after IV administration. Only a weak toxicity was observed in healthy tissues.	[48]
Non phospholipid-based nanoparticles			
Paccal Vet	In vitro efficacy	Paccal Vet (paclitaxel-loaded micelles) decreased the viability of canine hemangiosarcoma cells.	[49]
Lipid nanocapsules functionalized with the NFL ^2^- peptide	In vitro efficacy	The NFL-peptide promoted a better uptake and cytotoxicity of lipid nanocapsules in J3T canine glioblastoma cells	[50]
Lipid based nanoparticles containing miR-124	In vivo safety	The formulation was demonstrated to be safe when IV administered in healthy beagles.	[51]

^1^ MTD = Maximum tolerated dose; ^2^ NFL= NeuroFilament Light.

**Table 2 jfb-13-00116-t002:** Types of cancer and site of metastasis occurring before the beginning of the treatment with lipid-based carriers containing antitumor drugs.

Drug Delivery Systems	Cancer Types	Reference
Liposomal doxorubicin	Multiple myeloma	[52]
Doxil	Mycosis fungoides; anal gland adenocarcinoma; non-Hodgkin’s lymphoma; Malignant melanoma; mammary gland carcinoma; hemangiosarcoma; squamous cell carcinoma; thymoma; mast cell tumor; anaplastic sarcoma; malignant histiocytoma; fibrosarcoma; transitional cell carcinoma; thyroid carcinoma; mesenchymoma; neurofibrosarcoma; pulmonary adenocarcinoma; sweat gland adenocarcinoma; multiple myeloma	[53]
Doxil	Splenic hemangiosarcoma	[54]
Doxil	Splenic hemangiosarcoma.	[55]
Doxil	non-Hodgkin’s lymphoma	[57]
Liposomes containing cisplatin	Osteosarcoma	[58]
Liposomes containing untargeted tumor RNA	Malignant glioma	[60]
LDC-containing canine endostatin	Cutaneous soft tissue sarcomas	[61]
Liposomal muramyl tripeptide-phosphatidylethanolamine (L-MTP-PE)	Hemangiosarcoma; osteosarcoma.	[62]
L-MTP-PE	Osteosarcoma	[63]
L-MTP-PE	Hemangiosarcoma	[64]
L-MTP-PE	Mammary carcinoma	[65]
L-MTP-PE	Oral melanoma	[66]
Phosphatidylcholine-based liposomes containing clodronate	Malignant histiocytosis; lung and adrenal glands metastasis	[68]
Lipocurc	Primary or metastatic pulmonary neoplasia	[69]
Non phospholipids-based nanoparticles		
Paccal Vet	Advanced stage mast cell tumor	[76]
Paccal Vet	Mast cell tumor; mammary tumor; lymphoma; squamous cell carcinoma; anal sac carcinoma; bladder transitional cell carcinoma; fibrosarcoma; hemangiosarcoma; histiocytoma; malignant melanoma; mediastinal mass; osteosarcoma; synovial cell sarcoma	[77]
Lipid nanoemulsions containing carmustine	Lymphoma	[80]

**Table 3 jfb-13-00116-t003:** In vivo and in vitro studies based on non-lipid carriers containing antitumor drugs.

Drug Delivery Systems	Study Type	Results	Reference
Paclitaxel-loaded gelatin nanoparticles	Pharmacokinetics	Gelatin nanoparticles promoted a three-fold greater concentration of paclitaxel in bladder tissues with respect to the free form of the drug.	[81]
Convention-enhanced delivery of cetuximab conjugated to iron-oxide nanoparticles	Pharmacokinetics and safety	Distribution volume of cetuximab-free and cetuximab-conjugated to iron-oxide nanoparticles (IONPs) was similar after CED ^1^ administration in healthy beagles; a slower infusion showed a more uniform diffusion. Both formulations were safe.	[82]
Hyaluronan-cisplatin nanoconjugate	Pharmacokinetics	Hyaluronan-cisplatin nanoconjugate intratumorally injected in five tumor-bearing dogs, dramatically increased the concentration of the active compound inside the tumor masses compared with the free form of the drug. In addition, a significant localization of cisplatin within sentinel lymph nodes was obtained.	[83]
Paclitaxel-loaded gelatin nanoparticles	Pharmacokinetics	Paclitaxel-loaded nanoparticles (PNP) intra-vesically injected once a week in healthy and tumor bearing dogs favoured (i) a constant concentration of the drug in urine, (ii) a systemic distribution of only 1% of the injected dosage, (iii) a localization in the bladder tissue four times higher compared with free paclitaxel.	[84]
Hyaluronan-cisplatin nanoconjugates	Pharmacokinetics and safety	Hyaluronan-cisplatin nanoconjugates linked by N-Ac-Lys residue promoted an increased AUC of the drug in treated dogs and determined a Tmax of 6 h, much higher than that of the free form of the active compound. These in vivo features decreased the toxicity of cisplatin.	[85]
PZ4-decorated micelles made up of polyethylene glycol and cholic acid containing imaging agents, daunorubicin or paclitaxel	In vitro efficacy	PZ4-decorated micelles selectively targeted canine bladder cancer cells but not normal urothelial cells. PLZ4 increased the cytotoxicity of daunorubicin and the cellular uptake of micelles.	[86]
Aptamer-functionalized doxorubicin-Polylactide nanoconjugates	In vitro efficacy	Aptamer-functionalized doxorubicin- polylactide nanoconjugates incubated with canine hemangiosarcoma cells increased the intracellular localization of the drug and its toxicity with respect to the aptamer-free formulation	[87]
Poly(lactic-co-glycolic acid) (PLGA)-*block*(b)-PEG functionalized with triphenylphosphonium (TPP) cation nanoparticles containing cisplatin prodrug	In vitro efficacy and pharmacokinetics	The targeting of the mitochondria by PLGA-(b)-PEG-TPP-based nanoparticles containing the cisplatin prodrug (T-platin-M-NPs). The nanosystems significantly increased the toxicity of carboplatin and cisplatin on canine glioma and glioblastoma cells. In vivo studies demonstrated that T-platin-M-NPs are able to overcome the BBB ^2^ and reach the brain. T-platin-M-NPs were shown to be safe, and no severe adverse effects occurred on organs	[88]
Paclitaxel and curcumin encapsulated into PEG-coated mesoporous silica nanoparticles	In vitro efficacy	Paclitaxel and curcumin co-encapsulated into PEG-lipid-coated silica nanoparticles increased their cytotoxicity on canine breast cells	[89]
Cockleshell derived CaCO_3_ nanoparticles containing doxorubicin	Safety	Cockleshell derived CaCO_3_ nanoparticles promoted a decreased cardio- and nephrotoxicity of doxorubicin after injection in healthy dogs	[90]
Doxorubicin conjugated to glutathione-stabilized gold nanoparticles	In vitro efficacy	Doxorubicin conjugated to glutathione-stabilized gold nanoparticles showed a higher cytotoxicity of the drug on canine osteosarcoma cell lines with respect to the free form of the active compound.	[91]

^1^ CED = Convection-enhanced delivery; ^2^ BBB = Blood-brain barrier.

**Table 4 jfb-13-00116-t004:** Types of cancer and site of metastasis, occurring before the beginning of the treatment with non-lipid-based carriers containing antitumor drugs.

Drug Delivery Systems	Cancer Types	Reference
Polymeric hyaluronan cisplatin-nanoconjugate	Oral squamous cell carcinomas; nasal cancers; sarcoma; anal sac adenocarcinoma	[92]
HylaPlat	Oral squamous cell carcinomas; regional lymph node metastasis	[97]
Pam-Doxo-NPs	Osteosarcoma	[99]
PEG-PLA-PCL based nanoparticles, containing temozolomide-loaded superparamagnetic iron oxide	Glioblastoma; Anaplastic Astrocytoma; Cystic meningioma; High-grade astrocytoma	[102]
Iron oxide nanoparticles and/or virus plant nanoparticles	Oral melanoma	[106]
GA-AuNPs	Prostatic carcinoma; regional lymph nodes metastasis	[109]

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
