# Peer review of "Application of Biocompatible Drug Delivery Nanosystems for the Treatment of Naturally Occurring Cancer in Dogs"

_jfb, 2022, doi:10.3390/jfb13030116_

Round 1
Reviewer 1 Report
This is a very interesting publication on the use of nanoparticle-based anti-cancer drug formulations for the treatment of cancer in dogs. The canine model is a particularly interesting choice for the development of anti-cancer therapies. The results of studies on these mammals, among the available models, seem to be reasonably transferable to humans. A wide range of examples of therapies are cited, both in groups with induced cancer and in clinical cases. The review also cites examples of studies on canine-derived tissue cells. This broad overview gives a picture of the achievable effects of raising the biocompatibility of the therapies used, and describes both successful trials and those that failed. Unfortunately, a unified picture of improved therapeutic properties of drugs does not emerge from the cited literature; nevertheless, a number of interesting effects have been observed, such as reduced systemic toxicity, higher tissue concentrations or better drug distribution, but additional research is necessary to verify the possible improvement strategies.
The review is written in correct English and reads well. The topic is of interest to anyone interested in the development of nanoparticle-based cancer therapies. The reviewer verified the consistency of the cited results with those reported in the cited papers and found no factual errors.
'intravenous' abbreviation should be unified to IV (not i.v.).
Line 388 - that sentence starting with 'In this study...' is preceeded with the citation 84, which is not the one reviewed. It is infact 81, and it can be inferred from the text before, but it would be easier to read if the cited sentence at line 388 started with: 'In the mentioned study [81]...'
Reviewer 2 Report
The present review is very well written, complete and highly relevant.
As I had the opportunity to comment, it is my opinion that the current manuscript is a thorough and well-written review on the use of nanoparticle-based formulations for cancer treatment in dogs, and it will be informative and appealing for the small animal practitioner.
It represents, in my perspective, a point of view closer to clinical application and it does not shy away from pointing out the gap between experimental laboratory models, in vitro and in vivo, and effective and safe use in clinical practice and natural disease.
The fact is that in veterinary clinical practice, as in human medicine, the therapeutic approaches to certain types of cancer are limited in efficacy and toxicity is a major concern and limitation. Nanocarriers are cost-effective and implementable in everyday clinical practice, and likely part of the solution for the challenges posed in clinical cases with forms of cancer deemed untreatable or with very poor prognosis with current therapeutic approaches.
I suggest the inclusion of a table summarizing the different canine neoplasms (and if metastatized) in which nanoparticle-based formulations have been used, as developed in the manuscript.
Reviewer 3 Report
This review discusses the state of the art in the use of nanoparticles for the treatment of dogs’ tumors and the connection between canine tumor models and the development of innovative nanomedicine. This is an interesting topic, which provides some research directions for the application of nanotechnology in the field of medicine. However, there are some defects that need to be modified and improved according to the following comments:
1. Since this is a review paper, it will be better if you indicate the meaning of the abbreviations so it can be understood by wide variety of readers. For example, what is CED in table2? What is BBB in table2? Also, add the abbreviations to where they are first mentioned.
2. You should add more examples of non-phospholipid-based nanoparticles.
3. It is good to point out the limitations of each drug delivery system.
4. Regarding the references section, it is recommended to add several published papers, as many papers about drug delivery systems and naturally occurring cancers in dogs have been published recently.
5. The topic of the manuscript is good, but I suggest that you could check spelling and grammar through the whole paper. For example, "Unfortunately, the treatment resulted inefficacious". This sentence has obvious grammatical errors.
Reviewer 4 Report
1. Abstract require rewriting with background, aim, methodology and finding
2. introduction require more explanation about cancer and related research
3. explain more research on each type nano-formulation for cancer treatment
Round 2
Reviewer 4 Report
Accept`